# RULE: NEURAL-SYMBOLIC KNOWLEDGE GRAPH REASONING WITH RULE EMBEDDING

## ABSTRACT

Knowledge graph (KG) reasoning is an important problem for knowledge graphs. It predicts missing links by reasoning on existing facts. Knowledge graph embedding (KGE) is one of the most popular methods to address this problem. It embeds entities and relations into low-dimensional vectors and uses the learned entity/relation embeddings to predict missing facts. However, KGE only uses zeroth-order (propositional) logic to encode existing triplets (e.g., "Alice is Bob's wife."); it is unable to leverage first-order (predicate) logic to represent generally applicable logical **rules** (e.g., "$\forall x, y$: $x$ is $y$'s wife $\rightarrow$ $y$ is $x$'s husband"). On the other hand, traditional rule-based KG reasoning methods usually rely on hard logical rule inference, making it brittle and hardly competitive with KGE. In this paper, we propose RulE, a novel and principled framework to represent and model logical rules and triplets. RulE jointly represents entities, relations and logical rules in a unified embedding space. By learning an embedding for each logical rule, RulE can perform logical rule inference in a soft way and give a confidence score to each grounded rule, similar to how KGE gives each triplet a confidence score. Compared to KGE alone, RulE allows injecting prior logical rule information into the embedding space, which improves the generalization of knowledge graph embedding. Besides, the learned confidence scores of rules improve the logical rule inference process by softly controlling the contribution of each rule, which alleviates the brittleness of logic. We evaluate our method with link prediction tasks. Experimental results on multiple benchmark KGs demonstrate the effectiveness of RulE. `https://github.com/XiaojuanTang/RulE`

## 1 INTRODUCTION

Knowledge graphs (KGs) usually store millions of real-world facts and are used in a variety of applications, such as recommender systems (Wang et al., 2018), question answering (Bordes et al., 2014) and information retrieval (Xiong et al., 2017). Examples of knowledge graphs include Freebase (Bollacker et al., 2008), WordNet(Miller, 1995) and YAGO (Suchanek et al., 2007). They represent entities as nodes and relations among the entities as edges. Each edge encodes a fact in the form of a triplet (head entity, relation, tail entity). However, KGs are usually highly incomplete due to the limitedness of human knowledge and web corpora, as well as imperfect extraction algorithms. Knowledge graph reasoning, which predicts missing facts by reasoning on existing facts, has thus become a popular research area in Artificial Intelligence. There are two prominent lines of work in this area: knowledge graph embedding (KGE), which embeds all entities and relations into vectors, and rule-based KG reasoning, which uses logical rules to infer new facts.

Knowledge graph embedding (KGE) methods such as TransE (Bordes et al., 2013), ComplEx (Trouillon et al., 2016) and RotatE (Sun et al., 2019) have received significant attention due to their effectiveness and scalability. They embed entities and relations into low-dimensional vectors or matrices, preserving the inherent structure and latent semantic information. By computing the score of each triplet in the continuous space, KGE effectively estimates the plausibility of unobserved triplets so that reasoning of missing facts becomes feasible. However, KGE only uses zeroth-order (propositional) logic to encode existing facts (e.g., "Alice is Bob's wife."). It cannot leverage first-order (predicate) logic, which uses the universal quantifier to represent generally applicable logical

---

†Corresponding authors.

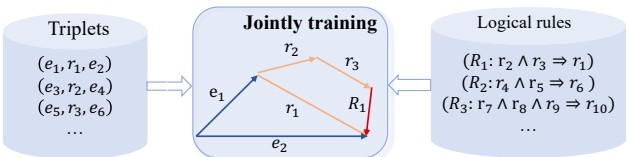

Figure 1: Given a KG containing a set of triplets and logical rules, RulE represents every entity, relation and logical rule as an embedding, i.e., $e, r, R$. It further defines respective mathematical operations between entities and relations as well as between relations and logical rules. The target is to jointly maximize the plausibility of existing triplets and logical rules, similar to how KGE maximizes the plausibility of observed triplets. In this way, we can optimize all embeddings $e, r, R$ in the same space to enhance each other, which improves the generalization of KGE and allows performing logical rule inference in a soft way.

rules. For instance, "$\forall x, y\colon x$ is $y$'s wife $\rightarrow y$ is $x$'s husband" encodes one logical rule. Those rules are not specific to particular entities (e.g., Alice and Bob) but are generally applicable to all entities. Another limitation of current KGE is that it can only be applied to transductive scenarios, which means it is unable to answer queries (h, r, t)s if t is not contained in KG during the training.

The other line of work, rule-based KG reasoning, in contrast, can be applied to both inductive and transductive settings, because rules are generally applicable to even unseen entities. Furthermore, unlike embedding-based methods, logical rules can achieve interpretable reasoning. However, the brittleness of logical rules greatly harms prediction performance. Consider the logical rule $(x, \text{work in}, y) \rightarrow (x, \text{live in}, y)$ as an example. It is mostly correct. Yet, if somebody works in New York but lives in New Jersey, the rule will surely infer the wrong fact that he/she lives in New York.

Considering that the aforementioned two lines of work can complement each other, addressing each other's weaknesses with their own forte, it becomes imperative to study how to robustly integrate logical rules with KGE methods. If we view this integration against a broader backdrop, embedding-based reasoning can be seen as a type of neural method, while rule-based reasoning can be seen as a type of symbolic method. Neural-symbolic synthesis has also been a focus of Artificial Intelligence research in recent years (Parisotto et al., 2017; Yi et al., 2018; Manhaeve et al., 2018; Xu et al., 2018; Hitzler, 2022). Specifically, most of the existing work combining logical rules and KGE either uses logical rules to infer new facts as additional training data for KGE or directly converts some rules into regularization terms for specific KGE models (Guo et al., 2016; Rocktäschel et al., 2015; Demeester et al., 2016). For example, KALE (Guo et al., 2016) and RUGE (Guo et al., 2018) apply t-norm fuzzy logic to grounded rules to give a truth score for each inferred triplet and generate additional triplets for KGE training. On the other hand, some other efforts do not ground rules to infer new triplets for training but rather inject logical rules via regularization terms on entity/relation embeddings during KGE training (Ding et al., 2018; Guo et al., 2020). They leverage logical rules merely to enhance KGE training without actually using logical rules to perform reasoning. In other words, they are still restricted to the transductive setting and might lose the important information contained in explicit rules, leading to empirically worse performance than state-of-the-art methods.

To address the aforementioned limitations, we propose *RulE*, a novel and principled framework to combine KGE and logical rules in a unified scheme. RulE stands for Rule Embedding. We choose it as our method's name to highlight that we are the first to jointly represent every entity, relation and logical rule in a unified embedding, as illustrated in Figure 1. Given a KG and a set of logical rules, RulE assigns an embedding vector to each entity, relation, and logical rule, and defines respective mathematical operators between entities and relations as well as between relations and logical rules. By jointly optimizing embeddings in the same space, RulE allows injecting prior logical rule information into the embedding space, which improves the generalization of KGE. Additionally, by learning an embedding for each logical rule, RulE is able to perform logical rule inference in a soft way as well as give a confidence score to each grounded rule, similar to how KGE gives each triplet a confidence score. The learned confidence scores can further improve the rule-based reasoning process by softly controlling the contribution of each rule, which alleviates the brittleness of logic.

We evaluate RulE on link prediction tasks and show superior performance. It performs competitively with strong baselines on knowledge graph completion benchmarks WN18RR (Dettmers et al., 2018)

and FB15K-237 (Toutanova & Chen, 2015), and obtains state-of-the-art performance on Kinship and UMLS (Kok & Domingos, 2007), two benchmark KGs for statistical relational learning.

To summarize, our main contributions in this work are the following. (1) We design a novel paradigm of neural-symbolic KG reasoning models to combine logical rules with KG embedding. To our best knowledge, this is the first attempt to embed entities, relations and logical rules into a unified space to perform KG reasoning. (2) Our framework is quite generic and flexible. It can adopt any KGE method, such as RotatE and TransE, and can use flexible ways to model the relationship between the embeddings of logical rules and their participating relations. (3) We empirically evaluate RulE on several benchmark KGs, showing superior performance to prior works.

## 2    RELATED WORK

A popular approach for knowledge graph reasoning is knowledge graph embedding (KGE), which embeds entities and relations into a continuous space and uses a scoring function to compute a score for each triplet based on the entity and relation embeddings (Bordes et al., 2013; Yang et al., 2014; Trouillon et al., 2016; Cai & Wang, 2017; Sun et al., 2019; Balažević et al., 2019; Vashishth et al., 2019; Zhang et al., 2020; Abboud et al., 2020). Much prior work in this regard views a relation as some operation or mapping function between entities. Most notably, TransE (Bordes et al., 2013) defines a relation as a translation operation between some head entity and tail entity. It is effective in modelling inverse and composition rules. DistMult (Yang et al., 2014) uses a bilinear mapping function to model symmetric patterns. RotatE (Sun et al., 2019) uses rotation operation in complex space to capture symmetry/antisymmetry, inversion and composition rules. **[BoxE (Abboud et al., 2020) models relations as boxes and entities as points to capture symmetry/anti-symmetry, inversion, hierarchy and intersection patterns but not composition rules]** Still, the relations well modelled by prior work are quite simple, and embeddings are learned solely based on triplets (zeroth-order logic) contained in the given KG. In contrast, our approach is able to embody more complex first-order logical rules in the embedding space by jointly modeling entities, relations and logical rules in a unified framework.

Besides embedding-based methods, logical rules have been widely applied in knowledge graph reasoning as well because of their interpretability and generalizability. As one of the early efforts, Quinlan (1990) uses Inductive Logic Programming (ILP) to derive logical rules (hypothesis) from all the training samples in a KG. Some later work leverages Markov Logic Networks (MLNs) to define the joint distribution over given (observed) and hidden variables (missing facts) (Kok & Domingos, 2005; Brocheler et al., 2012; Beltagy & Mooney, 2014). Using maximum likelihood estimation, MLNs learn the weights of logical rules, which can be further used to infer missing facts (hidden variables) in the probabilistic graph framework. However, this approach requires enumerating all possible facts and thus is not tractable. AMIE (Galárraga et al., 2013) and AMIE+ (Galárraga et al., 2015) first enumerate possible rules and then learn a scalar weight for each rule to encode the quality. Neural-LP (Yang et al., 2017) and DRUM (Sadeghian et al., 2019) mine rules by simultaneously learning logic rules and their weights based on TensorLog (Cohen et al., 2017). RNNLogic (Qu et al., 2020) simultaneously trains a rule generator as well as a reasoning predictor to generate high-quality logical rules. Except for RNNLogic, the above methods solely use the learned logical rules for reasoning, which suffer from brittleness and are hardly competitive with embedding-based reasoning in most benchmarks. Although RNNLogic considers the effect of KGE during inference, it only linearly combines the rule inference score and KGE score as the final prediction score, and pretrains KGE *separately* from logical rule learning without jointly modeling KGE and logical rules in the same space. In contrast, our RulE assigns an embedding to each logical rule and trains KGE together with rule embedding, so that they are modelled in the same embedding space to complement each other. In addition, during rule inference, RulE further employs the learned confidence scores of rules as soft multi-hot encoding of the activated rules instead of hard multi-hot encoding of RNNLogic to further promote rule-based reasoning.

Moreover, some recent work tries to incorporate logical rules into KGE models to improve the generalization and performance of KGE reasoning. KALE (Guo et al., 2016) applies t-norm fuzzy logic to grounded rules to give a truth score to each inferred triplet and generate additional triplets for KGE training. RUGE (Guo et al., 2018) additionally focuses on soft rules and uses logical rules to enhance KGE in an iterative manner. These models both use logical rules to infer new facts as additional training data for KGE. Besides the above ones, several other prior work injects

rules via regularization terms for KGE during training. Notably, Wang et al. (2015) formulates inference as an integer linear programming (ILP) problem with the objective function generated from embedding models and the constraints translated from rules. Ding et al. (2018) impose non-negativity constraints on entity representations by restricting each element of entity embedding to $[0, 1]$, and inject subrelation rules (e.g., $r_p \rightarrow r_q$) over relation representations, regularizing the score computed by KGE of the latter to be higher than that of the former. Based on Ding et al. (2018), Guo et al. (2020) go beyond simple rules (i.e., subrelation) and can handle more complex logical rules encoded by Horn clauses. In summary, these methods leverage logical rules only to enhance KGE training and do not really perform reasoning with logical rules. **[Besides, there are some other works that adopt graph neural networks framework. They usually aggregate local neighbor information to perform link prediction (Schlichtkrull et al., 2018; Teru et al., 2020; Zhu et al., 2021). Although they probably achieve better performance than our method in FB15k-237 and WN18RR, it does not explicitly apply logic rules and lack explainability. However, our method is more applicable paradigm such that we can exploit human domain knowledge.]** It simultaneously train KG embedding and rule embedding, and combine them together to infer missing facts, which mutually enhance each other.

## 3 PRELIMINARIES

A KG consists of a set of triplets or atoms $\mathcal{K} = \{(h, r, t) \mid h, t \in \mathcal{E}, r \in \mathcal{R}\} \subseteq \mathcal{E} \times \mathcal{R} \times \mathcal{E}$, where $\mathcal{E}$ denotes the set of entities and $\mathcal{R}$ the set of relations. For a testing triplet $(h, r, t)$, we define a query as $q = (h, r, ?)$. The knowledge graph reasoning (link prediction) task is to infer the missing entity t based on the existing facts (and rules).

### 3.1 EMBEDDING-BASED REASONING

Knowledge graph embedding (KGE) represents entities and relations in a continuous space called $embeddings$. It calculates a score for each triplet based on these embeddings via a scoring function. The embeddings are trained so that facts observed in the KG have higher scores than those not observed. The learning goal here is to maximize the scores of positive facts (existing facts) and minimize those of generated negative samples.

**RotatE** (Sun et al., 2019) is a representative KGE method with competitive performance on common benchmark datasets. It maps entities in a complex space and defines relations as element-wise rotations in a two-dimensional complex plane. Each entity and each relation is associated with a complex vector, i.e., $\boldsymbol{h}^{(c)}, \boldsymbol{r}^{(c)}, \boldsymbol{t}^{(c)} \in \mathbb{C}^k$, where the modulus of each element in $\boldsymbol{r}^{(c)}$ equals to 1 (multiplying a complex number with a unitary complex number is equivalent to a 2D rotation). The $k$-dimensional complex vector $\boldsymbol{h}_c$ can be expressed in the form $\boldsymbol{h}^{(c)} = \boldsymbol{h}_a + \boldsymbol{h}_b \boldsymbol{i}$, where $\boldsymbol{h}_a$ and $\boldsymbol{h}_b$ are $k$-dimensional real vectors. Based on it, RotateE represents each head entity as a $2k$-dimensional real vector $\boldsymbol{h} := [\boldsymbol{h}_a, \boldsymbol{h}_b] \in \mathbb{R}^{2k}$, where $\boldsymbol{h}_a$ are the real parts and $\boldsymbol{h}_b$ are the imaginary parts (the same for the tail embedding $\boldsymbol{t}$). Additionally, as rotations are associated with unitary complex numbers (i.e., $|[\boldsymbol{r}^{(c)}]_j| = 1, j = 1, 2, \ldots, k$), RotateE only parameterizes relation embeddings with $k$-dimensional real vectors (angles) $\boldsymbol{r}$. Here the real parts are $cos(\boldsymbol{r})$ and imaginary parts are $sin(\boldsymbol{r})$, where both $sin$ and $cos$ are element-wise operations.

If a triplet $(h, r, t)$ holds, it is expected that $\boldsymbol{t}^{(c)} \approx \boldsymbol{h}^{(c)} \circ \boldsymbol{r}^{(c)}$ in the complex space, where $\circ$ denotes the Hadamard (element-wise) product. This is equivalent to $\boldsymbol{t} \approx \text{Concat}\left(\boldsymbol{h}_a \circ cos(\boldsymbol{r}) - \boldsymbol{h}_b \circ sin(\boldsymbol{r}), \boldsymbol{h}_a \circ sin(\boldsymbol{r}) + \boldsymbol{h}_b \circ cos(\boldsymbol{r})\right)$ in the real space, where $\left(\boldsymbol{h}_a \circ cos(\boldsymbol{r}) - \boldsymbol{h}_b \circ sin(\boldsymbol{r})\right)$ is the real part of $\boldsymbol{t}$ and $\left(\boldsymbol{h}_a \circ sin(\boldsymbol{r}) + \boldsymbol{h}_b \circ cos(\boldsymbol{r})\right)$ is the imaginary part. The distance function of RotatE is defined as:

$$d(\boldsymbol{h}, \boldsymbol{r}, \boldsymbol{t}) = \| \text{Concat}\left(\boldsymbol{h}_a \circ cos(\boldsymbol{r}) - \boldsymbol{h}_b \circ sin(\boldsymbol{r}), \boldsymbol{h}_a \circ sin(\boldsymbol{r}) + \boldsymbol{h}_b \circ cos(\boldsymbol{r})\right) - \boldsymbol{t} \| \quad (1)$$

By defining relations as rotations in complex space, RotatE can model symmetry/antisymmetry, inversion and composition rules simultaneously.

### 3.2 RULE-BASED REASONING

Logical rules are usually expressed as first-order logic formulae, e.g., $\forall x, y, z \colon (x, r_1, y) \wedge (y, r_2, z) \rightarrow (x, r_3, z)$, or $r_1(x, y) \wedge r_2(y, z) \rightarrow r_3(x, z)$ for brevity. The left-hand side of the

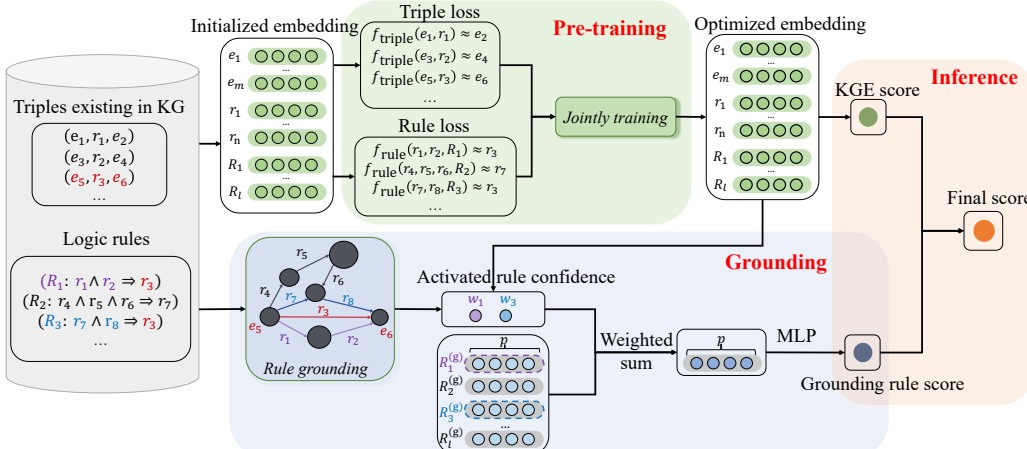

Figure 2: The overall architecture of our model. RulE consists of three components. Given a KG containing a set of triplets and logical rules extracted from the same or some other (external) knowledge source, we represent each entity, relation and logical rule as an embedding. 1) During the **pre-training** stage, we model the relationship between entities and relations as well as the relationship between relations and logical rules in the same continuous space to jointly learn entity, relation and rule embeddings. With the learned rule embeddings ($r$) and relation embeddings ($R$), RulE can output a weight ($w$) for each rule. 2) In the **grounding** stage, we further associate each rule with a learnable grounding-stage embedding ($R^{(g)}$). For all activated rules, we compute the weighted sum of their grounding-stage embeddings with $w$ which is further used to output a grounding rule score. 3) Finally, RulE integrates the KGE score obtained from the pre-training stage and the grounding rule score for **inference**.

implication "→" is called rule body or premise, and the right-hand side is rule head or conclusion. Logical rules are often restricted to be *closed*, forming *chains*. For a chain rule, successive relations share intermediate entities (e.g., $y$), and the conclusion's and premise's head/tail entity are the same. The length of a rule is the number of atoms (relations) that exist in its rule body. One example of a length-2 rule is:

$$\text{live\_in}(x, y) \wedge \text{city\_of}(y, z) \rightarrow \text{nationality}(x, z), \tag{2}$$

of which *live_in*(·) ∧ *city_of*(·) is the rule body and *nationality*(·) is the rule head. A *grounding* of a rule is obtained by substituting all variables $x, y, z$ with specific entities. For example, if we replace $x, y, z$ with *Bill Gates, Seattle, US* respectively, we get a grounding:

$$\text{live\_in}(\text{Bill Gates, Seattle}) \wedge \text{city\_of}(\text{Seattle, US}) \rightarrow \text{nationality\_of}(\text{Bill Gates, US}) \tag{3}$$

If all triplets in the rule body of a grounding exist in the KG, we get a *support* of this rule. Those rules that have nonzero support are called *activated* rules. When inferring a query $(\text{h}, \text{r}, ?)$, rule-based reasoning enumerates relation paths between head h and each candidate tail, and uses activated logic rules to infer the answer. For example, if we want to infer nationality(Bill Gates, ?), given the logical rule (2) as well as the existing triplets live_in(Bill Gates, Seattle) and city_of(Seattle, US), the answer US can be inferred by traversing the KG.

## 4 METHOD

This section introduces our proposed model RulE. RulE uses a novel paradigm to combine KG embedding with logical rules by learning rule embeddings. As illustrated in Figure 2, RulE consists of three key components. Consider a KG containing triplets and a set of logical rules extracted from the same or some other (external) knowledge source. 1) **Pretraining.** We model the relationship between entities and relations as well as the relationship between relations and logical rules to jointly train entity embedding, relation embedding and rule embedding in a continuous space, as illustrated in Figure 1. 2) **Grounding**. With the rule and relation embeddings, we calculate a confidence score for each rule which is used as the weight of activated rules to output a grounding rule score 3) **Inference**. Finally, we integrate the KGE score obtained from the pre-training stage and the grounding rule score obtainable from the grounding stage to reason unknown triplets.

## 4.1 PRE-TRAINING

Given a triplet $(h, r, t) \in \mathcal{K}$ and a rule $R \in \mathcal{L}$, we use $\boldsymbol{h}, \boldsymbol{r}, \boldsymbol{t}, \boldsymbol{R} \in \mathbb{R}^{2k}$ to represent their embeddings, respectively, where $k$ is the dimension of the complex space (following RotatE). Similar to KGE, which encodes the plausibility of each triplet with a scoring function, RulE additionally defines a scoring function for logical rules. Based on the two scoring functions, it jointly learns entity, relation and rule embeddings in a same space by maximizing the plausibility of existing triplets $\mathcal{K}$ (zeroth-order logic) and logical rules $\mathcal{L}$ (first-order logic). The following describes in detail how to model the triplets and logical rules together.

**Modeling the relationship between entities and relations** To model triplets, we take RotatE (Sun et al., 2019) due to its simplicity and competitive performance. Specifically, given a triplet $(h, r, t)$, its score is formulated as:

$$s_t(\boldsymbol{h}, \boldsymbol{r}, \boldsymbol{t}) = \gamma_t - d(\boldsymbol{h}, \boldsymbol{r}, \boldsymbol{t}), \tag{4}$$

where $\gamma_t$ is a fixed triplet margin and $d(\boldsymbol{h}, \boldsymbol{r}, \boldsymbol{t})$ is defined in Eq. (1). A triplet with a small distance will have a higher score, indicating its higher probability to be true. Note that our model is not restricted to RotatE. The distance function can also be replaced with other KGE models, such as TransE (Bordes et al., 2013) and ComplEx (Trouillon et al., 2016). Based on the scoring function, we define the loss function with negative sampling as:

$$L(\boldsymbol{h}, \boldsymbol{r}, \boldsymbol{t}) = -\log \sigma(s_t(\boldsymbol{h}, \boldsymbol{r}, \boldsymbol{t})) - \sum_{(\boldsymbol{h}', \boldsymbol{r}, \boldsymbol{t}') \in \mathbb{N}} \frac{1}{|\mathbb{N}|} \log \sigma(-s_t(\boldsymbol{h}', \boldsymbol{r}, \boldsymbol{t}')) \tag{5}$$

where $\sigma(x) = 1/(1 + \exp(-x))$ denotes the sigmoid function and $\mathbb{N}$ is the set of negative samples constructed by replacing either the head entity or the tail entity with a random entity (but not both at the same time). Following RotatE (Sun et al., 2019), we use a self-adversarial negative sampling approach to sample negative triplets from the current embedding model.

**Modeling the relationship between relations and logical rules** A universal first-order logical rule is some rule that universally holds for all entities. Therefore, we cannot relate such a rule with specific entities. Instead, it is a higher-level concept related to the relations of which it is composed. Motivated by RotatE, a natural idea is to model the relationship between relations and logical rule as element-wise rotations from the rule body to the rule head, i.e., for a logical rule $R: r_1 \wedge r_2 \wedge ... \wedge r_l \rightarrow r_{l+1}$, we expect that $\boldsymbol{r}_{l+1}^{(c)} \approx (\boldsymbol{r}_1^{(c)} \circ \boldsymbol{r}_2^{(c)} \circ ... \circ \boldsymbol{r}_l^{(c)}) \circ \boldsymbol{R}^{(c)}$, where $\{\boldsymbol{r}_i^{(c)}\}_{i=1}^l$ and $\boldsymbol{R}^{(c)}$ are in the $k$-dimensional complex space. However, 2D rotations are commutative—they cannot model the non-commutative property of composition rules, which is crucial for correctly expressing the relation order of a rule. Take sister_of$(x, y) \wedge$ mother_of$(y, z) \rightarrow$ aunt_of$(x, z)$ as an example. If we permute the relations in rule body, e.g., change (sister_of $\wedge$ mother_of) to (mother_of $\wedge$ sister_of), the rule is no longer correct. However, the above model will output the same score since $(\boldsymbol{r}_1^{(c)} \circ \boldsymbol{r}_2^{(c)}) = (\boldsymbol{r}_2^{(c)} \circ \boldsymbol{r}_1^{(c)})$.

Therefore, to respect the relation order of logical rules, we propose to use RNN (Graves, 2012). Given the sequence of all relations in a rule body, e.g., $[r_1, r_2, ..., r_l]$, we use RNN to generate the rule head $r_{l+1}$. Specifically, we first concatenate each relation embedding $\boldsymbol{r}_i$ in the rule body with the rule embedding $\boldsymbol{R}$. We further pass the concatenated embedding to an RNN, whose output is expected to be close to the rule head embedding $\boldsymbol{r}_{l+1}$. Given so, the distance function is formulated as:

$$d(\boldsymbol{R}, \boldsymbol{r}_1, \boldsymbol{r}_2, ..., \boldsymbol{r}_{l+1}) = \| \text{RNN}([\boldsymbol{R}, \boldsymbol{r}_1], [\boldsymbol{R}, \boldsymbol{r}_2], ..., [\boldsymbol{R}, \boldsymbol{r}_l]) - \boldsymbol{r}_{l+1} \|. \tag{6}$$

We also employ negative sampling, the same as when modeling triplets. At this time, it replaces a relation (either in rule body or rule head) with a random relation. The loss function for logical rules is defined as:

$$L(\boldsymbol{R}, \boldsymbol{r}_1, \boldsymbol{r}_2, ..., \boldsymbol{r}_{l+1}) = -\log \sigma(\gamma_r - d(\boldsymbol{R}, \boldsymbol{r}_1, \boldsymbol{r}_2, ..., \boldsymbol{r}_{l+1}))$$
$$- \sum_{(\boldsymbol{R}, \boldsymbol{r}_1', \boldsymbol{r}_2', ..., \boldsymbol{r}_{l+1}') \in \mathbb{M}} \frac{1}{|\mathbb{M}|} \log \sigma(d(\boldsymbol{R}, \boldsymbol{r}_1', \boldsymbol{r}_2', ..., \boldsymbol{r}_{l+1}') - \gamma_r), \tag{7}$$

where $\gamma_r$ is a fixed rule margin and $\mathbb{M}$ is the set of negative rule samples.

**Jointly training**  Given a KG containing a set of triplets $\mathcal{K}$ and logical rules $\mathcal{L}$, we jointly optimize the entity, relation, and rule embeddings with the loss:

$$L = \sum_{(\boldsymbol{h},\boldsymbol{r},\boldsymbol{t}) \in \mathcal{K}} L(\boldsymbol{h},\boldsymbol{r},\boldsymbol{t}) + \alpha \sum_{(\boldsymbol{R},\boldsymbol{r}_1,\boldsymbol{r}_2,...,\boldsymbol{r}_l) \in \mathcal{L}} L(\boldsymbol{R},\boldsymbol{r}_1,\boldsymbol{r}_2,...,\boldsymbol{r}_{l+1}), \tag{8}$$

where $\alpha$ is a positive hyperparameter weighting the importance of rules.

## 4.2 Grounding

During grounding, we use the parameters optimized by the pre-training process to compute the confidence score of each rule. Similar to how KGE gives a triplet the score, the confidence score of a logical rule $\mathrm{R}_i \colon \mathrm{r}_{i_1} \wedge \mathrm{r}_{i_2} \wedge ... \wedge \mathrm{r}_{i_l} \to \mathrm{r}_{i_{l+1}}$ is calculated by:

$$w_i = \gamma_r - d(\boldsymbol{R}_i, \boldsymbol{r}_{i_1}, \boldsymbol{r}_{i_2}, ..., \boldsymbol{r}_{i_{l+1}}) \tag{9}$$

where $d(\boldsymbol{R}_i, \boldsymbol{r}_{i_1}, \boldsymbol{r}_{i_2}, ..., \boldsymbol{r}_{i_l+1})$ is defined in Eq. (6). Each logical rule is further associated with a *grounding-stage rule embedding* $\boldsymbol{R}^{(g)} \in \mathbb{R}^p$. Note that this embedding is trained separately from the rule embedding $\boldsymbol{R}$ in the pre-training stage, and is only used for the grounding stage rule-based reasoning. Specifically, when answering a query $(\mathrm{h}, \mathrm{r}, ?)$, we apply logical rules to find different grounding paths on the KG, yielding different candidate answers. For each candidate answer $\mathrm{t}'$, we sum over the grounding-stage embeddings of those activated rules, each weighted by its rule confidence score $w_i$ and the number of paths activating this rule (# supports). Finally, we apply an MLP to calculate the grounding rule score:

$$s_g(\mathrm{h}, \mathrm{r}, \mathrm{t}') = \mathrm{MLP}\left( \mathrm{LN}(\sum_{\mathrm{R}_i \in \mathcal{L}'} w_i |\mathcal{P}(\mathrm{h}, \mathrm{R}_i, \mathrm{t}')| \boldsymbol{R}_i^{(g)}) \right) \tag{10}$$

where LN is the layer normalization operation, $\mathcal{L}'$ is the set of activated rules for $(\mathrm{h}, \mathrm{r}, \mathrm{t}')$, and $\mathcal{P}(\mathrm{h}, \mathrm{R}_i, \mathrm{t}')$ is the set of supports of the rule $\mathrm{R}_i$ which starts from h and ends at $\mathrm{t}'$. More implementation details of the grounding process are included in Appendix A. Once we have the grounding rule score for each candidate answer, we further use a softmax function to compute the probability of the true answer. Finally, we train the MLP and grounding-stage rule embeddings by maximizing the log likelihood of the true answers in the training triplets.

## 4.3 Inference

After pre-training and grounding, we predict any missing fact jointly with the KGE score (Eq. (4)) and the grounding rule score (Eq. (10)). Specifically, for a query $(\mathrm{h}, \mathrm{r}, ?)$, we substitute the tail entity with all candidate entities $(\mathrm{h}, \mathrm{r}, \mathrm{t}')$ and compute their KGE scores $s_t(\boldsymbol{h}, \boldsymbol{r}, \boldsymbol{t}')$. In addition, we perform a BFS search from h to find all candidates whose paths from h can activate a rule with the rule head grounded by $(\mathrm{h}, \mathrm{r}, \mathrm{t}')$. For these candidates, we compute their grounding rule scores defined in Eq. (10) (those activating no rules have zero grounding rule scores). Then the final scores of all candidate answers are computed by a weighted sum of $s_t(\boldsymbol{h}, \boldsymbol{r}, \boldsymbol{t}')$ and $s_g(\mathrm{h}, \mathrm{r}, \mathrm{t}')$:

$$s(\boldsymbol{h}, \boldsymbol{r}, \boldsymbol{t}') = s_t(\boldsymbol{h}, \boldsymbol{r}, \boldsymbol{t}') + \beta \cdot s_g(\mathrm{h}, \mathrm{r}, \mathrm{t}') \tag{11}$$

$\beta$ is a hyperparameter balancing the importance of embedding-based and rule-based reasoning.

## 5 Experiments

In this section, we experimentally evaluate RulE on several benchmark KGs and demonstrate the effectiveness of RulE on knowledge graph reasoning tasks.

## 5.1 Experiment settings

**Datasets.**  We choose four datasets for evaluation: FB15k-237 (Toutanova & Chen, 2015), WN18RR (Dettmers et al., 2018), UMLS and Kinship (Kok & Domingos, 2007). FB15k-237 and WN18RR are subsets of two large-scale knowledge graphs, FreeBase (Bollacker et al., 2008) and WordNet (Miller, 1995). UMLS and Kinship are two benchmark datasets for statistical relational learning. To ensure a fair comparison, we use the same splits as RNNLogic (Qu et al., 2020). More details of dataset statistics are summarized in Appendix C.

Table 1: Results of reasoning on UMLS and Kinship. H@k is in %. [*] means the numbers are taken from the **[RNNLogic paper]**. [†] means we rerun the methods with the same evaluation process.

| | UMLS | | | | Kinship | | | |
|---|---|---|---|---|---|---|---|---|
| | MRR | H@1 | H@3 | H@10 | MRR | H@1 | H@3 | H@10 |
| TransE† | 0.629 | 36.4 | 88.4 | 96.1 | 0.251 | 1.62 | 37.8 | 72.8 |
| DistMult* | 0.391 | 25.6 | 44.5 | 66.9 | 0.354 | 18.9 | 40.0 | 75.5 |
| ComplEx* | 0.411 | 27.3 | 46.8 | 70.0 | 0.418 | 24.2 | 49.9 | 81.2 |
| TuckER* | 0.732 | 62.5 | 81.2 | 90.9 | 0.603 | 46.2 | 69.8 | 86.3 |
| RotatE* | 0.744 | 63.6 | 82.2 | 93.9 | 0.651 | 50.4 | 75.5 | 93.2 |
| MLN* | 0.688 | 58.7 | 75.5 | 86.9 | 0.351 | 18.9 | 40.8 | 70.7 |
| PathRank* | 0.197 | 14.8 | 21.4 | 25.2 | 0.369 | 27.2 | 41.6 | 67.3 |
| Neural-LP* | 0.483 | 33.2 | 56.3 | 77.5 | 0.302 | 16.7 | 33.9 | 59.6 |
| DRUM* | 0.548 | 35.8 | 69.9 | 85.4 | 0.334 | 18.3 | 37.8 | 67.5 |
| RNNLogic+ (*rule.*)† | 0.800 | 70.4 | 87.8 | 94.3 | 0.655 | 50.4 | 76.0 | 94.7 |
| RNNLogic+ (*emb.+rule.*)† | 0.847 | 76.7 | 91.6 | 96.9 | 0.714 | 58.1 | 81.8 | 95.4 |
| RulE (*emb.*) | 0.801 | 70.0 | 88.2 | 96.0 | 0.674 | 53.6 | 76.6 | 93.7 |
| RulE (*rule.*) | 0.827 | 74.6 | 88.9 | 95.7 | 0.682 | 53.5 | 78.9 | 95.1 |
| RulE (*emb.+rule.*) | **0.866** | **79.5** | **92.7** | **97.4** | **0.740** | **62.0** | **82.9** | **95.7** |

Table 2: Results of reasoning on FB15k-237 and WN18RR. H@k is in %. [*] means the numbers are taken from the original papers. [†] means we rerun the methods with the same evaluation process.

| | FB15k-237 | | | | WN18RR | | | |
|---|---|---|---|---|---|---|---|---|
| | MRR | H@1 | H@3 | H@10 | MRR | H@1 | H@3 | H@10 |
| TransE* | 0.294 | - | - | 46.5 | 0.226 | - | - | 50.1 |
| DistMult* | 0.241 | 15.5 | 26.3 | 41.9 | 0.43 | 39 | 44 | 49 |
| ComplEx* | 0.247 | 15.8 | 27.5 | 42.8 | 0.44 | 41 | 46 | 51 |
| ComplEx-N3* | 0.37 | - | - | 56 | 0.48 | - | - | 57 |
| ConvE* | 0.325 | 23.7 | 35.6 | 50.1 | 0.43 | 40 | 44 | 52 |
| TuckER* | **0.358** | **26.6** | **39.4** | **54.4** | 0.470 | 44.3 | 48.2 | 52.6 |
| RotatE* | 0.338 | 24.1 | 37.5 | 53.3 | 0.476 | 42.8 | 49.2 | 57.1 |
| PathRank* | 0.087 | 7.4 | 9.2 | 11.2 | 0.189 | 17.1 | 20.0 | 22.5 |
| Neural-LP* | 0.237 | 17.3 | 25.9 | 36.1 | 0.381 | 36.8 | 38.6 | 40.8 |
| DRUM* | 0.238 | 17.4 | 26.1 | 36.4 | 0.382 | 36.9 | 38.8 | 41.0 |
| RNNLogic+ (*rule.*)† | 0.299 | 21.5 | 32.8 | 46.4 | 0.489 | 45.3 | 50.6 | 56.3 |
| RNNLogic+ (*emb.+rule.*)† | 0.349 | 25.8 | 38.5 | 53.3 | 0.502 | 46.0 | 51.8 | 58.1 |
| RulE (*emb.*) | 0.338 | 24.1 | 37.6 | 53.3 | 0.480 | 43.7 | 49.6 | 56.1 |
| RulE (*rule.*) | 0.311 | 22.9 | 34.1 | 47.6 | 0.491 | 45.4 | 50.7 | 56.1 |
| RulE (*emb.+rule.*) | 0.354 | 26.1 | 39.1 | 54.3 | **0.506** | **46.6** | **52.2** | **58.9** |

**Baselines.** We compare with a comprehensive suite of baselines. (1) *Embedding-based models*: we include TransE (Bordes et al., 2013), ComplEx (Trouillon et al., 2016), RotatE (Sun et al., 2019) and TuckER (Balažević et al., 2019). (2) *Rule-based models*: we consider popular rule-mining models Neural-LP (Yang et al., 2017), DRUM (Sadeghian et al., 2019), and two RNNLogic+ variants (Qu et al., 2020). RNNLogic+ (*rule.*) is a pure rule-based reasoning method while RNNLogic+ (*emb.+rule.*) uses both logical rules and knowledge graph embeddings. See more introductions to RNNLogic in Appendix B. Besides, we also compare with two early works: MLN (Richardson & Domingos, 2006) and PathRank (Lao & Cohen, 2010). (3) *RulE*: For our model RulE, we present results of embedding-based reasoning, rule-based reasoning and joint reasoning. The first one only uses KGE scores obtained from the pre-training stage to reason unknown triplets, denoted as (*emb.*). The second one uses the grounding score calculated from the grounding stage to infer missing facts, and we denote it as (*rule.*). The last combines both of them, denoted (*emb.+rule.*). By default, we use the same rules mined by RNNLogic+.

**Evaluation protocols.** For each test triplet $(h, r, t)$, we substitute the tail entity with all entities, calculate a score for each candidate, and sort all candidates in descending order to get the rank of the true tail entity. During ranking, we use the filtered setting (Bordes et al., 2013) by removing corrupted triplets that already exist in either the training, validation or test set. We report the standard evaluation metrics for these datasets, Mean Reciprocal Rank (MRR) and Hits at N (H@N).

Table 3: Results of reasoning on FB15k and WN18. H@k is in %.

| | **FB15k** | | | | **WN18** | | | |
|---|---|---|---|---|---|---|---|---|
| | MRR | H@1 | H@3 | H@10 | MRR | H@1 | H@3 | H@10 |
| TransE[†] | 0.730 | 64.6 | 79.2 | 86.4 | 0.772 | **70.5** | 80.8 | 92.2 |
| KALE | 0.523 | 38.3 | 61.6 | 76.2 | 0.662 | - | 85.5 | 93.0 |
| RulE (*emb. TransE*) | **0.734** | **65.0** | **79.9** | **86.9** | **0.775** | 67.2 | **86.2** | **95.0** |
| ComplEx[†] | 0.766 | 69.7 | 81.3 | 88.3 | 0.898 | 85.4 | 92.6 | **95.2** |
| RUGE | 0.768 | 70.3 | 81.5 | 86.5 | 0.943 | - | - | 94.4 |
| ComplEx-NNE+AER | **0.803** | **76.1** | 83.1 | 87.4 | **0.943** | **94.0** | **94.5** | 94.8 |
| RulE (*emb. ComplEx*) | 0.788 | 72.4 | **83.3** | **89.6** | 0.928 | 91.9 | 93.5 | 94.4 |

**Hyperparameter settings.** By default, we use RotatE (Sun et al., 2019) as our KGE model. We optimize our model parameters with Adam (Kingma & Ba, 2014) and tune the hyperparameters by grid search on the validation dataset. See more details in Appendix C.

## 5.2 RESULTS

As shown in Table 1 and 2, RulE outperforms all baselines on UMLS, Kinship, and WN18RR. It is competitive on FB15k-237 as well. Especially for UMLS and Kinship, we obtain 1.9% and 2.6% higher absolute MRR than the best baselines, respectively. A detailed analysis follows.

**Embedding logical rules helps KGE.** We first compare RulE (*emb.*) with RotatE. Compared to RotatE, RulE (*emb.*) only adds an additional rule embedding loss to the KGE training and still uses KGE scores only for prediction. As presented in Table 1 and Table 2, RulE (*emb.*) achieves higher performance than RotatE, indicating that embedding entities, relations and logical rules in the same space are beneficial for learning more compact representations for KG. Specifically, on UMLS and Kinship, RulE (*emb.*) outperforms RotatE with 5.7% and 2.3% improvement in MRR, which is more significant than on FB15k-237 and WN18RR. The reason is probably that UMLS and Kinship contain more rule-sensitive facts while WN18RR and FB15k-237 consist of more general facts (like the publication year of an album, which is hard to infer via rules). This phenomenon is observed in previous works too (Qu et al., 2020). In Table 3, we further experiment with variants of RulE (*emb.*) using TransE and ComplEx as the KGE models. They both obtain superior performance to the corresponding KGE models. More results are shown in Appendix E.1.

**Rule confidence improves rule-based reasoning.** One significant difference between RulE (*rule.*) and RNNLogic+ (*rule.*) is that the former weights the grounding-stage rule embeddings (Eq. (9)) when performing rule-based reasoning, while the latter only uses hard 1/0 to select activated rules. RulE (*rule.*) achieves better performance than RNNLogic+ (*rule.*). This demonstrates that the confidence scores of logical rules, which are learned through jointly embedding KG and logical rules, help rule-based reasoning. See Appendix E.2 for more discussion.

**Comparison with other joint reasoning and rule-enhanced KGE models.** Compared with RNNLogic+ (*emb.+rule.*), RulE (*emb.+rule.*) achieves better results on all datasets, demonstrating the benefits of pretraining KGE with rule embeddings together. We further compare RulE (*emb.*) with KALE (Guo et al., 2016), RUGE (Guo et al., 2018) and ComplEx-NNE+AER (Ding et al., 2018) in Table 3. These models inject logical rules to enhance KGE training. KALE is based on TransE, whereas RUGE and ComplEx-NNE+AER use ComplEx. For a fair comparison, we replace RotatE in RulE with TransE and ComplEx, respectively. As these baselines only have code for old datasets, we compare all models on FB15k and WN18. From Table 3, we can see that RulE (*emb. TransE*) yields more accurate results than KALE. For RulE (*emb. ComplEx*), although it does not outperform ComplEx-NNE+AER (probably because it additional injects the regularization terms on entities but RulE does not), compared to RUGE, RulE (*emb. ComplEx*) also obtains 2% improvement in MRR on FB15k as well as comparable results on WN18.

## 6 DISCUSSION AND CONCLUSION

We propose a novel and principled framework RulE to jointly represent entities, relations and logical rules in a unified embedding space. It improves the generalization capability of KGE and improves logical rule reasoning through absorbing the best of both. However, similar to prior work using logical rules, RulE needs to enumerate all paths between entity pairs, making it difficult to scale. In the future, we plan to explore more efficient and effective algorithms, e.g., replace logical rules completely with rule embeddings during inference, and consider more complex rules.

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

## A  MORE DETAILS OF IMPLEMENTING THE GROUNDING PROCESS

This section introduces how to implement the grounding process. It is slightly different from our proposed framework but can empirically achieve better results. Recall that $\boldsymbol{R}$ and $\boldsymbol{r}$ are the embeddings of logical rules and relations, respectively, whose dimensions are $k$. In practice, we do not need to restrict the dimension of $\boldsymbol{R}$ as $k$. With the relation embeddings and rule embeddings optimized in the pre-training process, we can compute the confidence score of each rule. Specifically, given a logical rule $R_i : r_{i_1} \wedge r_{i_2} \wedge ... \wedge r_{i_l} \to r_{i_{l+1}}$, the learned RNN outputs a derived embedding:

$$\boldsymbol{r}'_{i_{l+1}} = \mathrm{RNN}([\boldsymbol{R}_i, \boldsymbol{r}_{i_1}], [\boldsymbol{R}_i, \boldsymbol{r}_{i_2}], ..., [\boldsymbol{R}_i, \boldsymbol{r}_{i_l}]), \tag{12}$$

where the dimension of $\boldsymbol{r}'_{i_{l+1}}$ is $k$, the same as the rule head embedding $\boldsymbol{r}_{i_{l+1}}$. The distance between $\boldsymbol{r}'_{i_{l+1}}$ and $\boldsymbol{r}_{i_{l+1}}$ measures the uncertainty of the logical rule, thus the confidence function of $R_i$ is formulated as:

$$\boldsymbol{c}_i = \frac{\gamma_r}{k} - (\boldsymbol{r}'_{i_{l+1}} - \boldsymbol{r}_{i_{l+1}})^2, \tag{13}$$

where $\gamma_r$ is the fixed rule margin defined in the pre-training process, and $\boldsymbol{c}_i$ is a $k$-dimensional vector. Each element of $\boldsymbol{c}_i$ represents a way of weighting the rule $R_i$. Instead of the scalar $w_i$ defined in Eq. (9), we use a more fine-grained way to represent the confidence of each logical rule. When answering a query $(\mathrm{h}, \mathrm{r}, ?)$, we ground logical rules on the KG and yield different candidate answers. For each candidate answer $\mathrm{t}'$, we can define the following function:

$$\boldsymbol{v}_j(\mathrm{h}, \mathrm{r}, \mathrm{t}') = \mathrm{LN}\Big(\sum_{R_i \in \mathcal{L}'} c_{i,j} |\mathcal{P}(\mathrm{h}, R_i, \mathrm{t}')| \boldsymbol{R}_i^{(g)}\Big), j = 1, 2, \ldots, k \tag{14}$$

where LN is the layer normalization operation, $\mathcal{L}'$ is the set of logical rules activated by $(\mathrm{h}, \mathrm{r}, \mathrm{t}')$, $\boldsymbol{R}_i^{(g)}$ is the grounding-stage rule embedding of $R_i$ and $\mathcal{P}(\mathrm{h}, R_i, \mathrm{t}')$ is the set of supports of the rule $R_i$ which starts from h and ends at $\mathrm{t}'$. Eq. (14) encodes a way to activate the grounding-stage rule embedding with rule confidence. There are $k$ ways in total; thus, we can aggregate them to calculate the grounding score, i.e.,

$$s_{\mathrm{agg}}(\mathrm{h}, \mathrm{r}, \mathrm{t}') = \mathrm{MLP}(\mathrm{AGG}(\boldsymbol{v}_1(\mathrm{h}, \mathrm{r}, \mathrm{t}'), \boldsymbol{v}_2(\mathrm{h}, \mathrm{r}, \mathrm{t}'), \ldots, \boldsymbol{v}_k(\mathrm{h}, \mathrm{r}, \mathrm{t}'))) \tag{15}$$

Here, AGG is an aggregator, which can aggregate all $\boldsymbol{v}_j(\mathrm{h}, \mathrm{r}, \mathrm{t}')$ by sum, mean, max or min. In experiments, we adopt the sum in WN18RR and FB15k-237 as well as the mean in UMLS and Kinship. Once we have the grounding rule score $s_{\mathrm{agg}}(\mathrm{h}, \mathrm{r}, \mathrm{t}')$ for each candidate answer, we further compute the probability of the true answer by a softmax function. Finally, we optimize the MLP and grounding-stage rule embedding by maximizing the log likelihood of the true answers based on these training triplets.

In fact, if we aggregate $\boldsymbol{v}_j$ by sum and remove the layer normalization operation, Eq. (15) can be reduce to Eq. (10). The proof is shown as follows:

**Proof:**  We remove the layer normalization and rewrite Eq. (14) and Eq. (10):

$$\boldsymbol{v}_j(\mathrm{h}, \mathrm{r}, \mathrm{t}') = \sum_{R_i \in \mathcal{L}'} c_{i,j} |\mathcal{P}(\mathrm{h}, R_i, \mathrm{t}')| \boldsymbol{R}_i^{(g)} \tag{16}$$

$$s'_g(\mathrm{h}, \mathrm{r}, \mathrm{t}') = \mathrm{MLP}\Big(\sum_{R_i \in \mathcal{L}'} w_i |\mathcal{P}(\mathrm{h}, R_i, \mathrm{t}')| \boldsymbol{R}_i^{(g)}\Big) \tag{17}$$

By aggregating $\boldsymbol{v}_j$ by sum and combining Eq. (14) and Eq. (15), we obtain:

$$s'_{\text{agg}}(h, r, t') = \text{MLP}\left(\sum_{j=1}^{k} \boldsymbol{v}_j(h, r, t')\right)$$

$$= \text{MLP}\left(\sum_{j=1}^{k} \sum_{R_i \in \mathcal{L}'} c_{i,j} |\mathcal{P}(h, R_i, t')| \boldsymbol{R}_i^{(g)}\right)$$

$$= \text{MLP}\left(\sum_{R_i \in \mathcal{L}'} \sum_{j=1}^{k} c_{i,j} |\mathcal{P}(h, R_i, t')| \boldsymbol{R}_i^{(g)}\right) \qquad (18)$$

$$= \text{MLP}\left(\sum_{R_i \in \mathcal{L}'} |\mathcal{P}(h, R_i, t')| \boldsymbol{R}_i^{(g)} \sum_{j=1}^{k} c_{i,j}\right)$$

Based on Eq. (9), we imply that:

$$\sum_{j=1}^{k} c_{i,j} = \sum_{j=1}^{k} [\frac{\gamma_r}{k} - (\boldsymbol{r}'_{i_{l+1}} - \boldsymbol{r}_{i_{l+1}})^2]_j$$

$$= \gamma_r - \| \boldsymbol{r}'_{i_{l+1}} - \boldsymbol{r}_{i_{l+1}} \| \qquad (19)$$

$$= \gamma_r - \| \text{RNN}([\boldsymbol{R}_i, \boldsymbol{r}_{i_1}], [\boldsymbol{R}_i, \boldsymbol{r}_{i_2}], ..., [\boldsymbol{R}_i, \boldsymbol{r}_{i_l}]) - \boldsymbol{r}_{i_{l+1}} \|$$

$$= w_i$$

Thus we continue to simplify Eq. (18):

$$s'_{\text{agg}}(h, r, t') = \text{MLP}\left(\sum_{R_i \in \mathcal{L}'} |\mathcal{P}(h, R_i, t')| \boldsymbol{R}_i^{(g)} \sum_{j=1}^{k} c_{i,j}\right)$$

$$= \text{MLP}\left(\sum_{R_i \in \mathcal{L}'} w_i |\mathcal{P}(h, R_i, t')| \boldsymbol{R}_i^{(g)}\right) \qquad (20)$$

$$= s'_g(h, r, t')$$

This completes the proof.

## B  INTRODUCTION OF RNNLOGIC+

RNNLogic (Qu et al., 2020) aims to learn logical rules from knowledge graphs, which simultaneously trains a rule generator as well as a reasoning predictor. The former is used to generate rules while the latter learns the confidence of generated rules. RulE applies pre-defined logical rules to perform knowledge graph reasoning. To compare with it, we only focus on the reasoning predictor RNNLogic+, which is also a more powerful model than RNNLogic for exploiting logical rules. Its details are described in this section.

Given a KG containing a set of triplets and logical rules, RNNlogic+ associates each logical rule with a grounding-stage rule embedding $\boldsymbol{R}^{(g)}$ (following the grounding process of RulE), for a query $(h, r, ?)$, it grounds logical rules into the KG, finding different candidate answers. For each candidate answer $t'$, RNNLogic+ aggregates all the rule embeddings of those activated rules, each weighted by the number of paths activating this rule (# supports). This is a different point from RulE that RulE additionally employs the confidence scores of rules as soft multi-hot encoding instead of the hard multi-hot encoding of RNNLogic. Then an MLP is further used to project the aggregated embedding to the grounding rule score:

$$s_r(h, r, t') = \text{MLP}\left(\text{LN}(\text{AGG}(\{\boldsymbol{R}_i^{(g)}, |\mathcal{P}(h, R_i, t')|\}_{R_i \in \mathcal{L}'}))\right) \qquad (21)$$

where LN is the layer normalization operation, AGG is the PNA aggregator (Corso et al., 2020), $\mathcal{L}'$ is the set of activated rules for $(h, r, t')$, and $\mathcal{P}(h, R_i, t')$ is the set of supports of the rule $R_i$ which starts from h and ends at $t'$. Once RNNLogic+ has the score of each candidate answer, it can use a softmax function to compute the probability of the true answer. Finally, it trains the MLP and rule embeddings by maximizing the log likelihood of the true answers based on training triplets.

During inference, there are two variants of models:

- RNNLogic+ (*rule.*): This variant only uses the logical rules for knowledge graph reasoning. Specifically, we calculate the score of each candidate answer only defined in Eq. 21 (i.e., $s_r(h, r, t')$).
- RNNLogic+ (*emb.+rule.*): It uses RotatE (Sun et al., 2019) to *pretrain* knowledge graph embeddings models, which is another difference from RulE that RulE jointly models KGE and logical rules in the same space to learn entity, relation and logical rule embeddings. In the inference process, the same as RulE, it linearly combines the grounding rule score and KGE score as the final prediction score, i.e.,

$$s(h, r, t') = s_r(h, r, t') + \alpha * \text{KGE}(h, r, t'), \tag{22}$$

where $\text{KGE}(h, r, t')$ is the KGE score calculated with entity and relation embeddings optimized by RotatE and $\alpha$ is a positive hyperparameter weighting the importance of the knowledge graph embedding score.

## C    EXPERIMENT SETUP

### C.1    DATA STATISTICS

[We use the same splits as RNNLogic and mine logic rules by RNNLogic. Each relation has about 100 or 200 logic rules, and the length of rules is no more than 3.] More statistics of datasets are shown in Table 4.

Table 4: Statistics of four datasets

| Dataset | #Entities | #Relations | #Train | #Validation | #Test | #Rules |
|---------|-----------|------------|--------|-------------|-------|--------|
| FB15k-237 | 14,541 | 237 | 272,115 | 17,535 | 20,466 | 47,362 |
| WN18RR | 40,943 | 11 | 86,835 | 3,034 | 3,134 | 6,676 |
| UMLS | 135 | 46 | 1,959 | 1,306 | 3,264 | 18,400 |
| Kinship | 104 | 25 | 3,206 | 2,137 | 5,343 | 10,000 |

### C.2    DATA PROCESS

Most rules mined by rule mining systems are not chain rules. They usually need to be transformed into chain rules by inversing some relations. Considering $r_1(x, y) \wedge r_2(x, z) \to r_3(y, z)$ as an example, with replacing $r_1(x, y)$ with $r_1^{-1}(y, x)$, the rule can be converted into chain rule $r_1(y, x)^{-1} \wedge r_2(x, z) \to r_3(y, z)$. Based on the above, for data processing, we need to add a inverse version triplet $(t, r^{-1}, h)$ for each triplet $(h, r, t)$, representing the inverse relationship $r^{-1}$ between entity t and entity h.

### C.3    HYPERPARAMETERS SETTING

For the pre-training of RulE, the ranges of the hyperparameters for the grid search are: embedding dimension $k \in \{100, 200, 500, 1000\}$, learning rate $lr \in \{0.00001, 0.00005, 0.0001\}$, batch size of triplets and rules $b \in \{256, 512, 1024\}$, negative sample size $b_n \in \{128, 256, 512\}$, self-adversarial sampling temperature $\alpha \in \{0.5, 1.0\}$ and fixed margin $\gamma_t, \gamma_r \in \{6, 12, 24, 32\}$. During the grounding process. We tune the learning rate $lr_g \in \{0.0005, 0.001, 0.05\}$, the dimension of grounding-state rule embedding $p \in \{100, 200, 500\}$. For inference, we combine embedding-based reasoning and rule-based reasoning with the hyperparameter $\alpha \in \{0.5, 1.0, 3.0, 5.0\}$.

## D    A VARIANT OF MODELING RULES

Besides using RNN (Graves, 2012) to model logical rules, we propose another variant to model the relationship between logical rules and relations, which is called *ADD*. Recall that we are given the sequence of all relations in the rule body, e.g., $[r_1, r_2, \dots, r_l]$, ADD aims to generate the rule

head. To respect the relation order of logical rules, we associate each rule with a rule embedding, $\boldsymbol{R} = [\boldsymbol{R}^1, \boldsymbol{R}^2, ..., \boldsymbol{R}^l], R \in \mathbb{R}^{kl}$, where $k$ is the dimension of relation embedding and $l$ is the length of the logical rule. Based on these definitions, we can formulate the distance between the rule body and rule head as:

$$d(\boldsymbol{r}_1, \boldsymbol{r}_2, \dots, \boldsymbol{r}_{l+1}, \boldsymbol{R}) = \| \sum_{k=1}^{l} (\boldsymbol{r}_k \circ \boldsymbol{R}^k) - \boldsymbol{r}_{l+1} \|, \tag{23}$$

where $\circ$ is an element-wise product. Then we use Eq. (7) to further define loss function of logical rules.

## E    ABLATION STUDY

### E.1    EMBEDDING LOGICAL RULES BASED ON TRANSE

We first compare RulE (*emb.*)  with KGE models. As shown in Table 3, the two variants using TransE (Bordes et al., 2013) and ComplEx (Trouillon et al., 2016) as KGE models are called RulE (*emb. TransE*) and RulE (*emb. ComplEx.*)  respectively. They both obtain superior performance to the pure corresponding KGE models. We also further compare with other rule-enhance KGE models. In experiment setup, RulE (*emb. TransE*) uses the same logical rules as KALE (Guo et al., 2016); RulE (*emb. ComplEx*) uses the same logical rules as ComplEx-NNE-AER (Ding et al., 2018). From the comparison, RulE (*emb. TransE*) and RulE (*emb. ComplEx*) both achieve better performance than KALE and RUGE, respectively. From Table 5, we also see that RulE (*emb. ComplEx*) is slightly worse than ComplEx-NNE-AER. The reason is probably that ComplEx-NNE-AER additionally injects the regularization terms on entity representations but RulE does not, and ComplEx-NNE-AER is based on origin ComplEx, which performs better than ComplEx we use.

We further study the effect made by logical rules on embedding-based models with more datasets (i.e., UMLS and Kinship). We rerun KALE (Guo et al., 2016) on UMLS and Kinship datasets. They both use the same splits and logical rules as RulE. However, KALE uses pairwise ranking loss function while TransE and RulE use logistic loss function. Many experiments indicate that logistic loss usually performs better than pairwise ranking loss. Thus the results of KALE are worse than TransE in Table 5. It is not easy to refactor due to their old coding. So we focus more on TransE, RulE (*emb. TransE_RNN*)and RulE (*emb. TransE_ADD*). From the results shown in Table 5, we can see that whatever RNN or ADD, RulE (*emb.*)  achieves better results than TransE. Especially for ADD, RulE (*emb.TransE_ADD*) obtained 13.9% and 14.2% MRR improvements on UMLS and Kinship respectively.

Table 5: Results of reasoning on UMLS and Kinship. H@k is in %. TransE is the the KGE model.

|  | UMLS | | | | Kinship | | | |
|---|---|---|---|---|---|---|---|---|
|  | MRR | H@1 | H@3 | H@10 | MRR | H@1 | H@3 | H@10 |
| TransE | 0.629 | 36.4 | **88.4** | **96.1** | 0.251 | 1.62 | 37.8 | 72.8 |
| KALE | 0.462 | 31.1 | 55 | 74.2 | 0.284 | 19.2 | 30.2 | 45.6 |
| RulE (*emb. TransE_ADD*) | **0.768** | **66.8** | 85.0 | 93.2 | **0.393** | **23.9** | **45.1** | **72.9** |
| RulE (*emb. TransE_RNN*) | 0.658 | 43.2 | 86.3 | 95.5 | 0.282 | 8.3 | 37.1 | 69.0 |

Table 6: Results of reasoning on UMLS and Kinship. H@k is in %.

|  | UMLS | | | | Kinship | | | |
|---|---|---|---|---|---|---|---|---|
|  | MRR | H@1 | H@3 | H@10 | MRR | H@1 | H@3 | H@10 |
| RulE (*w/o confidence*) | 0.798 | 70.5 | 87.2 | 94.5 | 0.653 | 50.5 | 75.6 | 93.0 |
| RulE (*with confidence*) | **0.824** | **74.3** | **88.8** | **95.3** | **0.682** | **53.5** | **78.9** | **95.1** |

### E.2    RULE CONFIDENCE IMPROVE RULE-BASED REASONING

To fully verify our conclusion that rule confidence can promote rule-based reasoning, we focus on the grounding stage and remove the rule confidence, (i.e. assignment each $w_i$ with 1 on Eq. (9)),

which denoted it as RulE (*w/o confidence*). Both RulE (*w/o confidence*) and RulE (*with confidence*) use the same hyperparameters and tune them based on the validation sets. From Table 6, we can see that RulE (*with confidence*) produce more accurate results than RulE (*w/o* confidence), which demonstrates our the effectiveness of soft multi-hot encoding for grounding-stage rule embeddings.

