# OpenReview forum: "RulE: Neural-Symbolic Knowledge Graph Reasoning with Rule Embedding"
_ICLR.cc/2023/Conference — Submitted to ICLR 2023_

### Official Review · Reviewer_w9mu · 2022-10-19

**Confidence:** 4
**Correctness:** 2
**Technical Novelty And Significance:** 3
**Empirical Novelty And Significance:** 2
**Recommendation:** 3

**Clarity, Quality, Novelty And Reproducibility:**

## Clarity
Overall, this paper is clearly written. However, the authors spend too much space introducing properties of existing methods. For instance, on Page 6, the example that discusses the commutativity of RotatE can be removed.

## Quality
The proposed approach is overall reasonable. The experimental results are not significant and some results are unconvincing.

## Novelty
The idea of learning rules and graph embeddings simultaneously is not new, but the way to realize it in a unified space seems interesting.

## Reproducibility
Detailed experimental settings and codes are provided, so the reproducibility looks nice.

**Strength And Weaknesses:**

## Strength
1. The idea of embedding both knowledge graphs and logical rules in a unified space and learning them simultaneously is reasonable.
2. The preliminaries are well introduced and related works are fully discussed.
3. Detailed experimental settings and codes are provided for reproducibility.

## Weakness
1. **Motivation**
At the top of Page 2, the authors say that one of the limitations of current KGE is that they can only be used in transductive scenarios, and rule-based methods do not have this shortcoming. However, it seems that the proposed RulE does not resolve this limitation and is still transductive.
2. **Method**
The authors may want to analyze the complexity of RulE. Using RNNs to encode paths is time-consuming, especially when there are a lot of rules in a knowledge graph.
3. **Experiments**
My major concern is about the experimental results.
3.1 Note that the authors claim that RulE is proposed for better knowledge graph completion (KGC). However, RulE does not show clear performance improvements on the commonly-used KGC datasets (WN18RR and FB15k-237), which demonstrates the poor applicability of RulE. I agree with the authors' analyses that this is because WN18RR and FB15k-237 consist of more general facts and the phenomenon is also observed in previous works. However, the aim of the previous works is to learn logical rules instead of improving KGC performance with logical rules.
3.2 In the caption of Table 1, the authors claim that [*] means the numbers are taken from the original papers. However, I cannot find the corresponding results in these papers (e.g., TuckER and RotatE). In my experience, KGE models such as RotatE perform very well on UMLS and Kinship, which is far greater than the reported results.

**Summary Of The Paper:**

This paper proposes a new knowledge graph reasoning (KGR) model that combines embedding-based and rule-based methods. The authors claim that this is the first attempt to embed entities, relations, and logical rules into a unified space for KGR.

**Summary Of The Review:**

This paper proposes an interesting idea, but the significance of experiments should be greatly improved before it can be accepted.

---

> ### Author Response · Authors · 2022-11-11
> **Response to w9mu**
>
>
> - **Motivation**
>
>     We would like to clarify that our method is not only adapted to transductive. If we remove the KGE score during inference, by applying logic rules explicitly, we can perform KG reasoning and generalize unknown entities.
>
> - **Method**
>
>     We use RNN to encode participating relations unrelated to specific entities. And the length of the logical rules we use is small (no more than 3). Therefore, the complexity of time during the pre-training process is not significant. As for the grounding process, most rule-based methods require grounding in the entire KG, resulting in a number of grounding paths and high complexity. As pointed out in the conclusion of this paper, we will address this issue in the future.
>
> - **Experiment:**
>
>     (1) As you pointed out, RulE does not show clear performance improvements on the WN18RR and FB15k-237 datasets, but in Kinship and UMLS datasets it significantly outperforms embedding-based methods. Our method mainly focuses on jointly training entity, relation and logic rule embedding in a unified framework, and by injecting the prior logic information into the embedding space to improve the generalization of KGE. To demonstrate the claim, we conduct additional experiments on a new dataset YAGO3-10, which is a larger dataset than FB15k-237. More results are provided in the response of all reviewers.
>
>     (2) Thanks for pointing it out. For table 1, we make a mistake in that the numbers of UMLS and Kinship are copied from the RNNLogic paper. For a fair comparison with RNNLogic, we use the same split of datasets, thus KGE models perform relatively poorly on UMLS and Kinship. We are sorry about our carelessness, and will further revise that in the caption of Table 1.

---

### Official Review · Reviewer_fam7 · 2022-10-21

**Confidence:** 5
**Correctness:** 4
**Technical Novelty And Significance:** 2
**Empirical Novelty And Significance:** 2
**Recommendation:** 3

**Clarity, Quality, Novelty And Reproducibility:**

# Clarity:
The main ideas in the paper are presented clearly.

# Quality:
The ideas proposed in this paper are technically correct, but introduce several undesirable effects, particularly for a model seeking to introduce symbolic structure into the KGE domain.

# Novelty:
RulE does introduce any particularly novel components: The use of an RNN to inject composition rules in an order-aware fashion is a minor contribution, and it unfortunately comes with multiple side-effects, as discussed earlier. Moreover, the grounding and inference with additional hyper-parameters are also components used in other works in the literature.

# Reproducibility:
No concerns about reproducibility.

**Strength And Weaknesses:**

# Strengths:
- The empirical performance of RulE is good.
- The intuition for representing rules in the same space as relations and entities is well-placed and a good inductive bias, though I have some concerns about the way it is done in this paper (see weaknesses below).

# Weaknesses:
- The paper claims that the way it includes rules is principled, in that all representations are in the same embedding space. This is technically correct, but is problematic in practice for many reasons. First, the semantics of the rule component are not compatible with those on the KGE side. That is, relation embeddings act as rotations in RotatE, but are used as RNN tokens without any observable semantics on the rule side, with no connection to their functional role in RotatE. Hence, this unity between representations does not provide any meaningful insights or constraints aside from the technical parameter sharing.

- Second, the rule incorporation conducted in this work is not interpretable, and offers no means to study the inductive capacity of the model. Indeed, the scoring of rule satisfaction depends on a black-box RNN model, which makes studying combinations of rules or the set of rules captured by RulE virtually impossible. As a model trying to inject the symbolic structure of rule embedding, the lack of interpretable semantics on the rule embedding side severely undermines the contribution of the work. To illustrate this point, the rules in RulE all pass through the RNN, but there is no means to assess which potentially hidden inference patterns can be/are captured in the resulting model, whereas this can be done (albeit to a limited extent) in plain RotatE. Hence, using RulE effectively costs KGE models their interpretability on the inference patterns side.

- Third, the set of rules being injected by RulE is restricted only to composition rules, and does not present a viable means to represent more general rules: Effectively, new components, e.g., other RNNs, must be introduced for different types of rules, which also loses the structure and semantics of said rules. In fact, it is not even clear how RulE can handle multiple sets of composition rules in a sound way, for instance by respecting potential deductive closures. For example, the rules r_1(x, y) ^ r2(y, z) -> r3(x, z) and r3(x, y) ^ r4(y, z) -> r5(x, z) imply a longer chain rule r_1(x, y) ^ r2(y, z) ^ r4(z, w) -> r5(x, w), but such a deduction cannot be guaranteed to hold, let alone read through the RNN representations of the rules.

- Fourth, the approach is based on a more sophisticated neural network architecture (an RNN) and introduces additional hyper-parameters to its base model. Moreover, it introduces a potentially very costly grounding step prior to inference. As a result, the RulE model appears problematic not only from a scalability perspective, and this is acknowledged by the authors, but also from a parametrization and tuning perspective. Indeed, RulE achieves slightly improved results, but at the cost of interpretability, respecting the semantics of logical rules, and a larger parametrization. This unfortunately makes this setup less convincing overall.

- On a different point, the paper presents its hybrid use of rule and entity/relation representation in the same space as a main contribution, and mentions related work along two categories: KGE models with potential rule regularization, and reasoning systems without embedding structure. However, it fails to mention the BoxE model (Abboud et al., 2020), which can inject a set of rules from a rule language naturally as part of its relation representations and enforce them interpretably and with guarantees. Granted, RulE allows for softer rule injection, where a rule can technically fail, whereas BoxE does not. However, RulE in return does not provide any guarantees, supports far less rules, does not study its own inductive capacity despite emphasizing the importance of encoding rules, and proposes a less unified treatment of relation representations that differs between the fact and rule levels.

[1] Ralph Abboud et al. BoxE: A Box Embedding Model for Knowledge Base Completion. NeurIPS 2020.

**Summary Of The Paper:**

The paper proposes RulE, a model for link prediction on knowledge graphs that jointly learns representations for entities, relations and rules in the embedding space. More specifically, RulE applies on top of the RotatE KGE model, and additionally includes a component for learning compositional rule embeddings: Given a composition rule r1(x_1, x_2) ^ r2(x_2, x_3) ... ^ r_k(x_k,+1) -> r_k+1(x_1, x_k+1), RulE applies an RNN over the relational representations of r_1, r_k to produce an order-specific representation for the rule.  This rule representation is then compared to r_k+1 as part of pre-training in the loss function as a separate term. Beyond pre-training, RulE uses grounding to establish weights for the rules provided to the model and compute a score for a fact based on the rules. This score then acts as a complementary measure to support the embedding-based score at inference time. Empirically, the model is evaluated on standard knowledge graph completion baselines and is shown to achieve competitive performance. Finally, the model is run on reasoning baselines UMLS and kinship and achieves state-of-the-art results.

**Summary Of The Review:**

Overall, the unification of entity, relation and rule embeddings in a single embedding is a desirable property. However, in RulE, this is done in a problematic fashion, using black-box components that introduce i) inconsistent and incompatible objects between fact-level and rule-level uses of relational embeddings, ii) sacrifice the interpretability of the input KGE model (if already interpretable) particularly with respect to the study of inference patterns, iii) can only enforce a limited type of rules (compositions), and provides no soundness guarantees, e.g., capturing deductive closures of rules, and iv) substantially increase the parametrization and complexity of KGE models for minimal empirical gain. The contributions of RulE are also not completely well-placed in their context, as other works such as BoxE inject richer rule languages with provable guarantees. Therefore, I currently lean toward rejecting the paper. However, I am happy to change my verdict should the authors address my above concerns.

---

> ### Author Response · Authors · 2022-11-11
> **Response to fam7**
>
> Thanks for your reviewing our paper and for the helpful comments.
>
> For your questions:
>
> (1) W1 and W2: At first, inspired by Figure 1, we would like to define logic rules as the rotation operation between rule body and rule head, similar to the rotation from the head entity to the tail entity. But then we found that it does not empirically perform well and does not satisfy the relation-order property. Considering that RNN can model sequence information and can fit any function based on the universal approximation theorem, including rotation and addition, we use it to model logic rules. And the usefulness of RNN for modeling logic rules is also verified in our experiments. Besides, we also use TransE as our KGE model and ADD to model the relationship between relations and logic rules. More details have been provided in Appendix E. The experimental results are presented in table 5, and we can see that using Add can achieve better performance than using RNN, this is probably because ADD is better at capturing the semantic information between relations and logical rules when using TransE as the KGE base model. In conclusion, these results demonstrate that jointly representing zeroth-order logic and first-order logic can improve the generalization of KGE.
>
> (2) W3: The logic rules used in RulE are not restricted only to composition rules. We have addressed the concern in reply to reviewer VpTQ W3.
>
> (3) W4: Most works that apply logic rules are time-consuming [1][2][3], and the grounding step does not the main contribution of our paper. In our approach, we emphasize that jointly training entity, relation and logic rule embedding in the same space can benefit in improving the generalization of KGE. Many experiments have been conducted to demonstrate our claims. Please refer to the summary in the reply to all reviewers.
>
> (4) W5: Thank you for pointing out the related work. We have cited and discussed the work in the updated submission.
>
>
>
>
> [1] Galárraga, Luis Antonio, et al. "AMIE: association rule mining under incomplete evidence in ontological knowledge bases." *Proceedings of the 22nd international conference on World Wide Web*. 2013.
>
> [2] Yang, Fan, Zhilin Yang, and William W. Cohen. "Differentiable learning of logical rules for knowledge base reasoning." *Advances in neural information processing systems* 30 (2017).
>
> [3] Sadeghian, Ali, et al. "Drum: End-to-end differentiable rule mining on knowledge graphs." *Advances in Neural Information Processing Systems* 32 (2019).

---

> > ### Comment · Reviewer_fam7 · 2022-11-19
> > **Reviewer Response**
> >
> > I thank the authors for their reply. Unfortunately, I remain unconvinced by the arguments made in response to my concerns. In particular, I am still concerned about the problematic nature of the injection of rules, such that it is not strictly following the semantics of the original scoring function. I appreciate the authors' reference to empirical results. However, my concern above was not due to empirical results, but rather out of principle, as I find that sacrificing the unified representation stemming from a scoring function is a major design limitation that appears counter-productive for injecting rules reliably. This is made more concerning given that a black-box RNN is involved. This RNN is indeed universal, but the negative aspect is that these models are hard to constrain and use to provide guarantees, which is the primary purpose of rule injection. I therefore ask the authors to please address this point from that perspective.
> >
> > Regarding the type of rules captured, I recommend that the authors clarify this in the paper, ideally with examples of how the model would look like in these use cases.

---

### Official Review · Reviewer_VpTQ · 2022-10-24

**Confidence:** 4
**Correctness:** 3
**Technical Novelty And Significance:** 2
**Empirical Novelty And Significance:** 2
**Recommendation:** 3

**Clarity, Quality, Novelty And Reproducibility:**

Understandable writing, but motivation/justification missing. Partly repeats unnecessary details (e.g., RotatE model), partly lacks important details (e.g., points D3/D4).

Quality OK. Novelty unclear.

Reproducibility unclear (D1).


**Strength And Weaknesses:**

Strengths:

S1. Relatively simple approach (but see W1)

S2. Some improvements over baselines (but see W1)

Weaknesses:

W1. Does not outperform simpler methods / falls behind SOTA. The paper, in its essence, combines the score of a KGE model with a score obtained from fired rules. This is done in a more direct fashion in [A], with similar or better results than presented in this paper. Also, the results are from from SOTA [B]. Both papers are not discussed; related work is not covered well.

W2. Approach not well justified. The paper presents its method clearly, but doesn't justify it. Personally, I am not convinced of the approach, perhaps for this reason. One point of concern is that the RNN, which produces the rule embeddings, is trained in a complete data-independent fashion (Eq. 7). Why such an approach would work better than augmenting the training process as done in prior work is unclear to me.

W3. Only chain rules supported.

W4. Study not insightful. The experimental study shows some results, but does not provide any additional insight. Where is it, exactly, that the proposed method can outperform alternatives?

Other points:

D1. The experimental study is silent on where the rules come from.

D2. The weight w_i of Eq. 9 doesn't appear to be a "confidence" in that it does not have a probabilistic interpretation.

D3. How is $R^(g)$ trained (Sec. 4.2)? Is it the output of the RNN? Or trained directly?

D4. The main paper describes a method for grounding, pointing to details in the appendix. The appendix describes a completely different method, however. This is highly confusing.

References:

[A] Meilicke et al., Why a Naive Way to Combine Symbolic and Latent Knowledge Base Completion Works Surprisingly Well. AKBC 2021

[B] Zhu et al., Neural Bellman-Ford Networks: A General Graph Neural Network Framework for Link Prediction. NeurIPS 2021


**Summary Of The Paper:**

Combines rule-based and embedding-based reasoning methods for link prediction in knowledge graphs. Key idea is to learn an embedding of chain rules via an RNN during KGE training. During inference, combines KGE score of target triple and rule-embedding score of rules that fire on it.

**Summary Of The Review:**

Provides improvements over baselines, but does not outperform simpler methods and is far from SOTA (both not cited/discussed).

---

> ### Author Response · Authors · 2022-11-11
> **Response 1/2**
>
> We thank reviewer VpTQ for the constructive review and comments to help us improve the paper.
>
> > W1. Does not outperform simpler methods / falls behind SOTA. The paper, in its essence, combines the score of a KGE model with a score obtained from fired rules. This is done in a more direct fashion in [A], with similar or better results than presented in this paper. Also, the results are from SOTA [B]. Both papers are not discussed; related work is not covered well.
> >
>
> (1) Here we would like to clarify that although the improvement of our experimental results is at the similar level to [A], by carefully checking it and we find that it uses a different baseline setting (i.e., using libKGE framework to implement KGE models). This is a general framework that can achieve better results in KGE. Thus, the comparison between this method and RulE is not fair. Moreover, in terms of the methodology, [A] only ensembles KGE methods and rule-based methods by setting proper hyper-parameters to balance the importance of two sides, but our method can enhance the KGE method by jointly training logical rules and KGE embeddings. You can refer to the reply to all reviewers for our contribution. In addition, we thank for you providing the framework libKGE and are also doing the experiment based on it to achieve superior performance.
>
> (2) Thank for you pointing out the related work [B]. We will add this discussion to the revised paper. Some preliminary discussion is as follows. In essence, NBFNet is a general graph neural network framework to solve KG reasoning. Although it achieves better performance than our method in FB15k-237 and WN18RR, it does not explicitly apply logic rules and lacks explainability. In contrast, our method is a more applicable paradigm such that we can exploit human domain knowledge. And in rule-sensitive datasets such as UMLS and Kinship, RulE also obtains significant performance and has certain interpretability.
>
> > W2. Approach not well justified. The paper presents its method clearly, but doesn't justify it. Personally, I am not convinced of the approach, perhaps for this reason. One point of concern is that the RNN, which produces the rule embeddings, is trained in a complete data-independent fashion (Eq. 7). Why such an approach would work better than augmenting the training process as done in prior work is unclear to me.
> >
>
> Firstly, we would like to clarify that RNN does not generate rule embeddings, but models the relationship between participating relations and logic rules. Please refer to Appendix E for more details. Secondly, a universal first-order logic rule is some rule that holds for all entities (i.e., independent of specific entities). Thus, when modeling a logical rule, we cannot relate the rules to specific triplets of KG. In practice, we have also tried to inject the frequency of grounding rules as weights of rule loss, but we found that it may dominate those logic rules that are important but do not frequently appear in the knowledge graph. Therefore, we finally used a data-independent approach to model the logic rules.
>
> > W3. Only chain rules supported.
> >
>
> Chain rules include common logic rules in knowledge graphs such as symmetry, inversion, composition, hierarchy, intersection rules, etc. We think these kinds of rules play a more important role in KG reasoning.
>
> > W4. The study is not insightful. The experimental study shows some results but does not provide any additional insight. Where is it, exactly, that the proposed method can outperform alternatives?
> >
>
> The intuition behind the idea is that jointly representing entities, relations and logic rules in a unified space can inject the semantics of logic rules into the embedding space such that it can improve the generalization of KGE and help learn more compatible rule embeddings. We provide insights in the reply to all reviewers and verify the usefulness of the components of RulE. You can also refer to Section 5 and Appendix E for more details.

---

> ### Author Response · Authors · 2022-11-11
> **Response 2/2**
>
> Other points:
> - Re D1: “The experimental study is silent on where the rules come from.”
>
>     We have mentioned that the rules are mined by RNNLogic+ in the section Baselines on page 8.
>
> - Re D2: “The weight $w_i$ of Eq. 9 doesn't appear to be a "confidence" in that it does not have a probabilistic interpretation.”
>
>     We are inspired by how KGE gives a triplet score, which measures the plausibility of the fact. The $w_i$ is computed in a similar way and encodes the uncertainty of each logic rule..
>
> - Re D3: “How is $R^{(g)}$ trained (Sec. 4.2)? Is it the output of the RNN? Or trained directly?”
>
>     We have addressed D3 in our response (2) to reviewer E2oH.
>
> - Re D4: “The main paper describes a method for grounding, pointing to details in the appendix. The appendix describes a completely different method, however. This is highly confusing.”
>
>     In the implementation of the grounding process, we use a slightly different method from our proposed theoretical method because we find that this method can empirically achieve better results. In fact, if we use a sum-aggregator and remove the layer normalization operation, the implementation method can be reduced to the theoretical method. Please refer to Appendix A proof.

---

### Official Review · Reviewer_E2oH · 2022-10-25

**Confidence:** 4
**Correctness:** 3
**Technical Novelty And Significance:** 3
**Empirical Novelty And Significance:** 3
**Recommendation:** 5

**Clarity, Quality, Novelty And Reproducibility:**

Overall the writing is good, except for the important details regarding grounding. The originality of the work is okay. However, the contribution is limited.


**Strength And Weaknesses:**

# Strength
The paper tackles an interesting task in knowledge graph embedding .The paper is generally well-written (but with some issues in clarity, see below).

# Weakness

There are some issues in the current format as detailed below:

1. There are several important aspects missing regarding logic and grounding, which are the core contribution of this work over previous ones. The exact training schema for grounding, including the training objects, are quite unclear in the paper. This makes it hard to judge the contribution of this paper.

2. There are some unclarity on the contribution of each part in the proposed method. The pre-training part, which is largely resembling RNNLogic (Qu et al.,), is pretty loosely composed with the grounding technique (a major contribution in this manuscript) in the inference. I'm afraid this only constitutes a limited contribution. Furthermore, it's not mentioned how $\beta$, the hyper-parameter to balance two parts, is selected.

3. This is some unclarity regarding the dataset. It's unclear what the distribution of first-order logic looks like (e.g. number of terms). This is important as such statistics would determine whether the proposed use of RNN is justified or there are better alternatives.


**Summary Of The Paper:**

This paper proposes to address the knowledge graph reason where, besides entity and relation representation, first order logic is also considered. In doing so the authors propose to enrich RotatE with a RNN for embedding score and a separate grounding embedding for scores.Two scores are shallowly composed in the inference time. The authors demonstrate the performance of the proposed method with the link prediction tasks


**Summary Of The Review:**

Based on strengths and weaknesses, I would recommend 5: marginally below the acceptance threshold.

---

> ### Author Response · Authors · 2022-11-11
> **Response to E2oH**
>
> We thank reviewer E2oH for the comments to help us improve the paper.
>
> There are three concerns in the review. Q1: the contribution of our work. Q2: the details of the grounding process. Q3: the experiment setting and datasets.
>
> **(1) The contribution of our work.**
>
> We have addressed the concern by summarizing our contribution in the reply to all reviewers.
>
> **(2) The details of the grounding process.**
>
> The implementation details of the grounding process have also been provided in Appendix A. You can refer to help you understand. Here we would like to explain them to you again.  During grounding, we use the parameters optimized in pre-training including relation and rule embeddings to compute the confidence score $w_i$ for each rule, where $i$ denotes the $i$-th rule. Then each rule is further associated with a *grounding-stage rule embedding* $R^{(g)}$, which is different from the rule embedding $R$ in the pre-training process and is optimized in the grounding process. For each triplet (in training dataset) $(h,r,t)$, we replace the tail entity with all candidate answers (e.g. $(h,r,t^\prime)$) and calculate the grounding rule score by aggregating those activated grounding-stage rule embeddings. Once we have the grounding rule score for each candidate answer, we further use a softmax function to compute the probability of the true answer (i.e., $\frac{exp(s_g(h,r,t))}{\sum_{t^\prime \in \mathcal E} exp(s_g(h,r,t^\prime))}$). Our training objective is to maximize the log-likelihood of the true answers in the training triplets by training the MLP and grounding-stage rule embeddings.
>
> **(3) The experiment setting and datasets.**
>
> Thanks for pointing it out, and we have updated it accordingly in the revised manuscript. The $\beta$  is the hyper-parameter to balance the importance of the embedding-based method and rule-based method according to the validation set. Specifically, we apply grid search for the best $\beta$ based on the filter MRR on the validation dataset, with the range of  {1, 1/3, 1/5}.  In practice, we find that setting $\beta$ as 1 or 1/3 usually achieves better performance. As for the statistics of logic rules, we mine logic rules by RNNLogic, and each relation has about 100 or 200 logic rules with a maximum length of 3.

---

### Author Response · Authors · 2022-11-11
**Response to all reviewers**

As pointed out by multiple reviewers, we would like to highlight that the main contribution of our work is that we devise a novel framework that closely connects KGE with logic rules. Most previous works separately train embedding-based methods and rule-based methods, then loosely ensemble the two methods. But in our work, we represent entities, relations and logic rules in the same space to jointly train zeroth-logic and first-order logic in a unified framework, as Figure 1 shown. This joint training has two advantages: **1) on the one hand**, jointly optimizing embeddings in the same space allows injecting prior logical rule information into the KGE embedding space, which improves the generalization of KGE; **2) on the other hand**, with the help of the optimized relation embeddings and rule embeddings, RulE can compute the confidence score for each logic rule, which enhances the original hard rule-based reasoning process through soft rule confidence. Finally, combining the jointly trained KGE and the confidence-enhanced rule-based reasoning, we arrive at a final neural-symbolic model achieving superior performance on many datasets.

To demonstrate the effectiveness of RulE, we have done extensive evaluations on benchmarks as well as several ablation studies to verify the usefulness of the components of RulE. These results are presented in Section 5 and Appendix E.

To verify 1), in Table 1 and Table 2, we used RotatE as the KGE method and RNN to model logical rules. Please refer to row 5 and row 12 of Table 1: RulE(emb.) outperforms RotatE, especially in UMLS with a 5.7% improvement in MRR. Note that RulE(emb.) only uses the KGE model (jointly trained with rule embedding) for inference, without really performing rule-based reasoning. This demonstrates that jointly training logical rules and KGE embeddings help KGE itself.

We have also done experiments with variants of RulE (emb.) using TransE and ComplEx as the KGE models. They both obtain superior performance to the corresponding KGE models. The results are provided in Table 3 and Table 5 in Appendix E. These results demonstrate our claim (i.e., jointly representing zeroth-order logic and first-order logic can improve the generalization of KGE.)

To further strengthen our argument, here we conduct additional experiments on a new dataset YAGO3-10, which is a larger dataset than FB15k-237. We still use the logical rules mined from RNNLogic. The prediction results  (MRR) are shown below:

| YAGO3-10 | MRR |
| --- | --- |
| RotatE | 49.5% |
| RulE (emb.) | 52.6% |

The results show that the joint training boosts KGE itself.

To verify 2), as presented in Table 1 and Table 2, RulE (rule.) achieves better performance than RNNLogic+ (rule.). Note that RulE (rule.) can be seen as enhancing RNNLogic+ (rule.) with the learned rule confidence in the pretraining stage (the latter method essentially uses the same confidence 1 for all rules). The improved performance indicates that the confidence scores of logical rules, which are learned by jointly embedding KG and logical rules, can help rule-based reasoning.

We believe that our contributions are clearly presented and sufficiently validated by the solid experimental results. Our model RulE, is the first model that can principally combine neural and symbolic methods through jointly embedding them in the same space. The joint embedding is never explored before, yet has demonstrated great potential as a principled method for unifying zeroth-order logic reasoning (entity/relation reasoning) and first-order logic reasoning (rule-based reasoning), benefitting both worlds as discussed in 1) and 2).

Please see our individual response to each reviewer below.

---

### Decision · Program_Chairs · 2023-01-20

**Decision:**

Reject

**Justification For Why Not Higher Score:**

Concerns about comparison to prior work, about the motivation of the approach and about novelty

**Justification For Why Not Lower Score:**

no lower score possible

**Metareview: Summary, Strengths And Weaknesses:**

SUMMARY

The paper combines rule-based and embedding-based reasoning
methods for link prediction in knowledge graphs. The key
idea is to learn an embedding of chain rules via an RNN
during KG embedding training. During inference, they combine
the KG embedding score of a target triple and rule-embedding
scores of rules that fire on it.

STRENGTHS

Interesting problem addressed

Some improvement over baselines

Paper is mostly written well

WEAKNESSES

Important related work is not discussed. The experimental
results of the paper are below those in this related
work. Some of the comparison figures from related work were
incorrect in the original submission.

Lack of motivation for the proposed approach (why should a
data-independent modeling approach be able to learn good
rule embeddings; this is also related to lack of
interpretability: in a blackbox approach it is not clear
what was learned)

The description of the proposed method in the original
submission is insufficient.

The description of the dataset in the original
submission is insufficient.

The paper combines two elements, one similar to previous
work, one novel one. It is not clear how much each part
contributes. That means it is not clear how much the novel
contribution of this paper contributes.

There is a suspicion that part of the good performance of
the model is due to the introduction of more parameters

Lack of analysis of results